# VIME: Extending the Success of Self- and Semi-supervised Learning to Tabular Domain

**Jinsung Yoon**
Google Cloud AI, UCLA
jinsungyoon@google.com

**Yao Zhang**
University of Cambridge
yz555@cam.ac.uk

**James Jordon**
University of Oxford
james.jordon@wolfson.ox.ac.uk

**Mihaela van der Schaar**
University of Cambridge
UCLA, Alan Turing Institute
mv472@cam.ac.uk

## Abstract

Self- and semi-supervised learning frameworks have made significant progress in training machine learning models with limited labeled data in image and language domains. These methods heavily rely on the unique structure in the domain datasets (such as spatial relationships in images or semantic relationships in language). They are not adaptable to general tabular data which does not have the same explicit structure as image and language data. In this paper, we fill this gap by proposing novel self- and semi-supervised learning frameworks for tabular data, which we refer to collectively as VIME (Value Imputation and Mask Estimation). We create a novel pretext task of estimating mask vectors from corrupted tabular data in addition to the reconstruction pretext task for self-supervised learning. We also introduce a novel tabular data augmentation method for self- and semi-supervised learning frameworks. In experiments, we evaluate the proposed framework in multiple tabular datasets from various application domains, such as genomics and clinical data. VIME exceeds state-of-the-art performance in comparison to the existing baseline methods.

## 1 Introduction

Tremendous successes have been achieved in a variety of applications (such as image classification [1], object detection [2], and language translation [3]) with deep learning models via supervised learning on large labeled datasets such as ImageNet [4]. Unfortunately, collecting sufficiently large labeled datasets is expensive and even impossible in several domains (such as medical datasets concerned with a particularly rare disease). In these settings, however, there is often a wealth of unlabeled data available - datasets are often collected from a large population, but target labels are only available for a small group of people. The 100,000 Genomes project [5], for instance, sequenced 100,000 genomes from around 85,000 NHS patients affected by a rare disease, such as cancer. By definition rare diseases occur in (less than) 1 in 2000 people. Datasets like these present huge opportunities for self- and semi-supervised learning algorithms, which can leverage the unlabeled data to further improve the performance of a predictive model.

Unfortunately, existing self- and semi-supervised learning algorithms are not effective for tabular data[1] because they heavily rely on the spatial or semantic structure of image or language data. A

standard self-supervised leaning framework designs a (set of) pretext task(s) to learn informative representations from the raw input features. For the language domain, BERT introduces 4 different pretext tasks (e.g. predicting future words from previous words) to learn representations of the language data [6]. In the image domain, rotation [7], jigsaw puzzle [8], and colorization [9] can be utilized as pretext tasks to learn representations of the images. Standard semi-supervised learning methods also suffer from the same problem, since the regularizers they use for the predictive model are based on some prior knowledge of these data structures. For example, the consistency regularizer encourages the predictive model to have the same output distribution on a sample and its augmented variants, e.g. an image and its rotated variants [7], or two images and their convex combination(s) [10]. The notion of rotation simply does not exist in tabular data. Moreover, in many settings, variables are often categorical, and do not admit meaningful convex combinations. Even in a setting where all variables are continuous, there is no guarantee that the data manifold is convex and as such taking convex combinations will either generate out-of-distribution samples (therefore degrading model performances) or be restricted to generating samples that are very close to real samples (limiting the effectiveness of the data augmentation), for more details see the Supplementary Materials (Section 4).

**Contribution:** In this paper, we propose novel self- and semi-supervised learning frameworks for tabular data. For self-supervised learning, we introduce a novel pretext task, *mask vector estimation* in addition to *feature vector estimation*. To solve those pretext tasks, an encoder function learns to construct informative representations from the raw features in the unlabeled data. For semi-supervised learning, we introduce a novel tabular data augmentation scheme. We use the trained encoder to generate multiple augmented samples for each data point by masking each point using several different masks and then imputing the corrupted values for each masked data point. Finally, we propose a systematic self- and semi-supervised learning framework for tabular data, VIME (Value Imputation and Mask Estimation), that combines our ideas to produce state-of-the-art performances on several tabular datasets with a few labeled samples, from various domains.

## 2 Related Works

**Self-supervised learning (Self-SL)** frameworks are representation learning methods using unlabeled data. It can be categorized into two types: using pretext task(s) and contrastive learning. Most existing works with pretext tasks are appropriate only for images or natural language: (i) surrogate classes prediction (scaling and translation) [11], (ii) rotation degree predictions [7], (iii) colorization [9], (iv) relative position of patches estimation [12], (v) jigsaw puzzle solving [8], (vi) image denoising [13], (vii) partial-to-partial registration [14], and (viii) next words and previous words predictions [6]. Most existing works with contrastive learning are also applicable only for image or natural languages due to their data augmentation scheme, and temporal and spatial relationships for defining the similarity: (i) contrastive predictive coding [15, 16], (ii) contrastive multi-view coding [17], (iii) SimCLR [18], (iv) momentum contrast [19, 20].

There is some existing work on self-supervised learning which can be applied to tabular data. In Denoising auto-encoder [21], the pretext task is to recover the original sample from a corrupted sample. In Context Encoder [22], the pretext task is to reconstruct the original sample from both the corrupted sample and the mask vector. The pretext task for self-supervised learning in TabNet [23] and TaBERT [24] is also recovering corrupted tabular data.

In this paper, we propose a new pretext task: to recover the mask vector, in addition to the original sample with a novel corrupted sample generation scheme. Also, we propose a novel tabular data augmentation scheme that can be combined with various contrastive learning frameworks to extend the self-supervised learning to tabular domains.

**Semi-supervised learning (Semi-SL)** frameworks can be categorized into two types: entropy minimization and consistency regularization. Entropy minimization encourages a classifier to output low entropy predictions on unlabeled data. For instance, [25] constructs hard labels from high-confidence predictions on unlabeled data, and train the network using these pseudo-labels together with labeled data in a supervised way. Consistency regularization encourages some sort of consistency between a sample and some stochastically altered version of itself. $\Pi$-model [26] uses an $L_2$ loss to encourage consistency between predictions. Mean teacher [27] uses an $L_2$ loss to encourage consistency between the intermediate representations. Virtual Adversarial Training (VAT) [28] encourages prediction consistency by minimizing the maximum difference in predictions between a sample and multiple

augmented versions. MixMatch [29] and ReMixMatch [30] combine entropy minimization with consistency regularization in one unified framework with MixUp [10] as the data augmentation method. There is a series of interesting works on graph-based semi-supervised learning [31, 32, 33] which consider a special case of network data where samples are connected by a given edge, i.e. a citation network where an article is connected with its citations. Here, we introduce a novel data augmentation method for general tabular data which can be combined with various semi-supervised learning frameworks to train a predictive model in a semi-supervised way.

## 3 Problem Formulation

In this section, we introduce the general formulation of self- and semi-supervised learning. Suppose we have a small labeled dataset $\mathcal{D}_l = \{\mathbf{x}_i, y_i\}_{i=1}^{N_l}$ and a large unlabeled dataset $\mathcal{D}_u = \{\mathbf{x}_i\}_{i=N_l+1}^{N_l+N_u}$, where $N_u \gg N_l$, $\mathbf{x}_i \in \mathcal{X} \subseteq \mathbb{R}^d$ and $y_i \in \mathcal{Y}$. The label $y_i$ is a scalar in single-task learning while it can be given as a multi-dimensional vector in multi-task learning. We assume every input feature $\mathbf{x}_i$ in $\mathcal{D}_l$ and $\mathcal{D}_u$ is sampled i.i.d. from a feature distribution $p_X$, and the labeled data pairs $(\mathbf{x}_i, y_i)$ in $\mathcal{D}_l$ are drawn from a joint distribution $p_{X,Y}$. When only limited labeled samples from $p_{X,Y}$ are available, a predictive model $f : \mathcal{X} \to \mathcal{Y}$ solely trained by supervised learning is likely to overfit the training samples since the empirical supervised loss $\sum_{i=1}^{N_l} l\big(f(\mathbf{x}_i), y_i\big)$ we minimize deviates significantly from the expected supervised loss $\mathbb{E}_{(\mathbf{x},y) \sim p_{X,Y}}\big[l\big(f(\mathbf{x}), y\big)\big]$, where $l(\cdot, \cdot)$ is some standard supervised loss function (e.g. cross-entropy).

### 3.1 Self-supervised learning

Self-supervised learning aims to learn informative representations from unlabeled data. In this subsection, we focus on self-supervised learning with various self-supervised/pretext tasks for a pretext model to solve. These tasks are set to be challenging but highly relevant to the downstream tasks that we attempt to solve. Ideally, the pretext model will extract some useful information from the raw data in the process of solving the pretext tasks. Then the extracted information can be utilized by the predictive model $f$ in the downstream tasks. In general, self-supervised learning constructs an encoder function $e : \mathcal{X} \to \mathcal{Z}$ that takes a sample $\mathbf{x} \in \mathcal{X}$ and returns an informative representation $\mathbf{z} = e(\mathbf{x}) \in \mathcal{Z}$. The representation $\mathbf{z}$ is optimized to solve a pretext task defined with a pseudo-label $y_s \in \mathcal{Y}_s$ and a self-supervised loss function $l_{ss}$. For example, the pretext task can be predicting the rotation degree of some rotated image in the raw dataset, where $y_s$ is the true rotation degree and $l_{ss}$ is the squared difference between the predicted rotation degree and $y_s$. We define the pretext predictive model as $h : \mathcal{Z} \to \mathcal{Y}_s$, which is trained jointly with the encoder function $e$ by minimizing the expected self-supervised loss function $l_{ss}$ as follows,

$$\min_{e,h} \mathbb{E}_{(\mathbf{x}_s, y_s) \sim p_{X_s, Y_s}}\Big[l_{ss}\big(y_s, (h \circ e)(\mathbf{x}_s)\big)\Big] \tag{1}$$

where $p_{X_s, Y_s}$ is a pretext distribution that generates pseudo-labeled samples $(\mathbf{x}_s, y_s)$ for training the encoder $e$ and pretext predictive model $h$. Note that we have sufficient samples to approximate the objective function above since for each input sample in $\mathcal{D}_u$, we can generate a pretext sample $(\mathbf{x}_s, y_s)$ for free, e.g. rotating an image $\mathbf{x}_i$ to create $\mathbf{x}_s$ and taking the rotation degree as the label $y_s$. After training, the encoder function $e$ can be used to extract better data representations from raw data for solving various downstream tasks. Note that in settings where the downstream task (and a loss for it) are known in advance, the encoder can be trained jointly with the downstream task's model.

### 3.2 Semi-supervised learning

Semi-supervised learning optimizes the predictive model $f$ by minimizing the supervised loss function jointly with some unsupervised loss function defined over the output space $\mathcal{Y}$. Formally, semi-supervised learning is formulated as an optimization problem as follows,

$$\min_f \mathbb{E}_{(\mathbf{x},y) \sim p_{XY}}\Big[l\big(y, f(\mathbf{x})\big)\Big] + \beta \cdot \mathbb{E}_{\mathbf{x} \sim p_X, \mathbf{x}' \sim \tilde{p}_X(\mathbf{x}'|\mathbf{x})}\Big[l_u\big(f(\mathbf{x}), f(\mathbf{x}')\big)\Big] \tag{2}$$

where $l_u : \mathcal{Y} \times \mathcal{Y} \to \mathbb{R}$ is an unsupervised loss function, and a hyperparameter $\beta \geq 0$ is introduced to control the trade-off between the supervised and unsupervised losses. $\mathbf{x}'$ is a perturbed version of $\mathbf{x}$ assumed to be drawn from a conditional distribution $\tilde{p}_X(\mathbf{x}'|\mathbf{x})$. The first term is estimated using

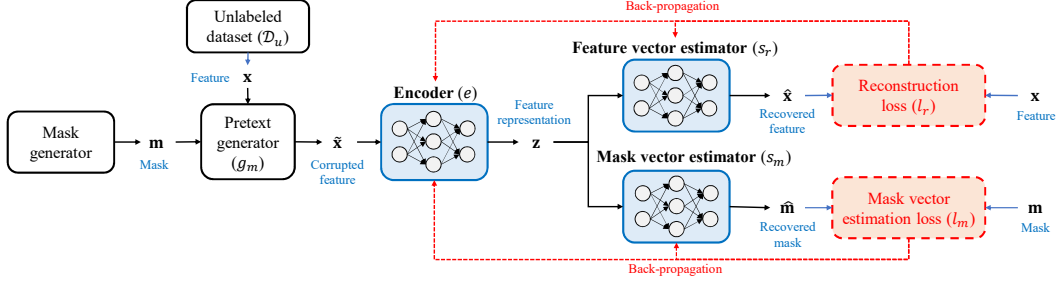

Figure 1: Block diagram of the proposed self-supervised learning framework on tabular data. (1) Mask generator generates binary mask vector ($\mathbf{m}$) which is combined with an input sample ($\mathbf{x}$) to create a masked and corrupted sample ($\tilde{\mathbf{x}}$), (2) Encoder ($e$) transforms $\tilde{\mathbf{x}}$ into a latent representation ($\mathbf{z}$), (3) Mask vector estimator ($s_m$) is trained by minimizing the cross-entropy loss with $\mathbf{m}$, feature vector estimator ($s_r$) is trained by minimizing the reconstruction loss with $\mathbf{x}$, (4) Encoder ($e$) is trained by minimizing the weighted sum of both losses.

the small labeled dataset $\mathcal{D}_l$, while the second term is estimated using all input features in $\mathcal{D}_u$. The unsupervised loss function ($l_u$) is often inspired by some prior knowledge of the downstream task. For example, consistency regularization encourages the model $f$ to produce the same output distribution when its inputs are perturbed ($\mathbf{x}'$).

## 4 Proposed Model: VIME

In this section, we describe VIME, our systematic approach for self- and semi-supervised learning for tabular data (block diagram can be found in the Supplementary Materials)). We first propose two pretext tasks in self-supervised learning, then we develop an unsupervised loss function in semi-supervised learning using the encoder learned from the pretext tasks via self-supervised learning.

### 4.1 Self-supervised learning for tabular data

We introduce two pretext tasks: *feature vector estimation* and *mask vector estimation*. Our goal is to optimize a pretext model to recover an input sample (a feature vector) from its corrupted variant, at the same time as estimating the mask vector that has been applied to the sample.

In our framework, the two pretext tasks share a single pretext distribution $p_{X_s, Y_s}$. First, a mask vector generator outputs a binary mask vector $\mathbf{m} = [m_1, ..., m_d]^\top \in \{0, 1\}^d$ where $m_j$ is randomly sampled from a Bernoulli distribution with probability $p_m$ (i.e. $p_\mathbf{m} = \prod_{j=1}^d \mathrm{Bern}(m_j|p_m)$). Then a pretext generator $g_m : \mathcal{X} \times \{0, 1\}^d \to \mathcal{X}$ takes a sample $\mathbf{x}$ from $\mathcal{D}_u$ and a mask vector $\mathbf{m}$ as input, and generates a masked sample $\tilde{\mathbf{x}}$. The generating process of $\tilde{\mathbf{x}}$ is given by

$$\tilde{\mathbf{x}} = g_m(\mathbf{x}, \mathbf{m}) = \mathbf{m} \odot \bar{\mathbf{x}} + (1 - \mathbf{m}) \odot \mathbf{x} \tag{3}$$

where the $j$-th feature of $\bar{\mathbf{x}}$ is sampled from the empirical distribution $\hat{p}_{X_j} = \frac{1}{N_u} \sum_{i=N_l+1}^{N_l+N_u} \delta(x_j = x_{i,j})$ where $x_{i,j}$ is the $j$-th feature of the $i$-th sample in $\mathcal{D}_u$ (i.e. the empirical marginal distribution of each feature). - see Figure 3 in the Supplementary Materials for further details. The generating process in Equation (3) ensures the corrupted sample $\tilde{\mathbf{x}}$ is not only tabular but also similar to the samples in $\mathcal{D}_u$. Compared with standard sample corruption approaches, e.g. adding Gaussian noise to, or replacing zeros with the missing features, our approach generates $\tilde{\mathbf{x}}$ that is more difficult to distinguish from $\mathbf{x}$. This difficulty is crucial for self-supervised learning, which we will elaborate more in the following sections.

There are two folds of randomness imposed in our pretext distribution $p_{X_s, Y_s}$. Explicitly, $\mathbf{m}$ is a random vector sampled from a Bernoulli distribution. Implicitly, the pretext generator $g_m$ is also a stochastic function whose randomness comes from $\bar{\mathbf{x}}$. Together, this randomness increases the difficulty in reconstructing $\mathbf{x}$ from $\tilde{\mathbf{x}}$. The level of difficulty can be adjusted by changing the hyperparameter $p_m$, the probability in $\mathrm{Bern}(\cdot|p_m)$, which controls the proportion of features that will be masked and corrupted.

Following the convention of self-supervised learning, the encoder $e$ first transforms the masked and corrupted sample $\tilde{\mathbf{x}}$ to a representation $\mathbf{z}$, then a pretext predictive model will be introduced to recover the original sample $\mathbf{x}$ from $\mathbf{z}$. Arguably, this is a more challenging task than existing pretext tasks, such as correcting the rotation of images or recoloring a grayscale image. A rotated or grayscale image still contains some information about the original features. In contrast, masking completely removes some of the features from $\mathbf{x}$ and replaces them with a noise sample $\bar{\mathbf{x}}$ of which each feature may come from a different random sample in $\mathcal{D}_u$. The resulting sample $\tilde{\mathbf{x}}$ may not contain any information about the missing features and even hard to identify which features are missing. To solve such a challenging task, we first divide it into two sub-tasks (pretext tasks):

(1) *Mask vector estimation*: predict *which* features have been masked;

(2) *Feature vector estimation*: predict the values of the features that have been corrupted.

We introduce a separate pretext predictive model for each pretext task. Both models operate on top of the representation $\mathbf{z}$ given by the encoder $e$ and try to estimate $\mathbf{m}$ and $\mathbf{x}$ collaboratively. The two models and their functions are,

- **Mask vector estimator**, $s_m : \mathcal{Z} \to [0,1]^d$, takes $\mathbf{z}$ as input and outputs a vector $\hat{\mathbf{m}}$ to predict which features of $\tilde{\mathbf{x}}$ have been replaced by a noisy counterpart (i.e., $\mathbf{m}$);
- **Feature vector estimator**, $s_r : \mathcal{Z} \to \mathcal{X}$, takes $\mathbf{z}$ as input and returns $\hat{\mathbf{x}}$, an estimate of the original sample $\mathbf{x}$.

The encoder $e$ and the pretext predictive models (in our case, the two estimators $s_m$ and $s_r$) are trained jointly in the following optimization problem,

$$\min_{e,s_m,s_r} \mathbb{E}_{\mathbf{x}\sim p_X, \mathbf{m}\sim p_{\mathbf{m}}, \tilde{\mathbf{x}}\sim g_m(\mathbf{x},\mathbf{m})} \Big[ l_m(\mathbf{m}, \hat{\mathbf{m}}) + \alpha \cdot l_r(\mathbf{x}, \hat{\mathbf{x}}) \Big] \tag{4}$$

where $\hat{\mathbf{m}} = (s_m \circ e)(\tilde{\mathbf{x}})$ and $\hat{\mathbf{x}} = (s_r \circ e)(\tilde{\mathbf{x}})$. The first loss function $l_m$ is the sum of the binary cross-entropy losses for each dimension of the mask vector[2]:

$$l_m(\mathbf{m}, \hat{\mathbf{m}}) = -\frac{1}{d} \Big[ \sum_{j=1}^{d} m_j \log \big[ (s_m \circ e)_j(\tilde{\mathbf{x}}) \big] + (1 - m_j) \log \big[ 1 - (s_m \circ e)_j(\tilde{\mathbf{x}}) \big] \Big], \tag{5}$$

and the second loss function $l_r$ is the reconstruction loss,

$$l_r(\mathbf{x}, \hat{\mathbf{x}}) = \frac{1}{d} \Big[ \sum_{j=1}^{d} (x_j - (s_r \circ e)_j(\tilde{\mathbf{x}}))^2 \Big]. \tag{6}$$

$\alpha$ adjusts the trade-off between the two losses. For categorical variables, we modified Equation 6 to cross-entropy loss. Figure 1 illustrates our entire self-supervised learning framework.

**What has the encoder learned?** These two loss functions share the encoder $e$. It is the only part we will utilize in the downstream tasks. To understand how the encoder is going to benefit these downstream tasks, we consider what the encoder must be able to do to solve our pretext tasks. We make the following intuitive observation: it is important for $e$ to capture *the correlation among the features* of $\mathbf{x}$ and output some latent representations $\mathbf{z}$ that can recover $\mathbf{x}$. In this case, $s_m$ can identify the masked features from the inconsistency between feature values, and $s_r$ can impute the masked features by learning from the correlated non-masked features. For instance, if the value of a feature is very different from its correlated features, this feature is likely masked and corrupted. We note that correlations are also learned in other self-supervised learning frameworks, e.g. spatial correlations in rotated images and autocorrelations between future and previous words. Our framework is novel in learning the correlations for tabular data whose correlation structure is less obvious than in images or language. The learned representation that captures the correlation across different parts of the object, regardless of the object type (e.g. language, image or tabular data), is an informative input for the various downstream tasks.

## 4.2 Semi-supervised learning for tabular data

We now show how the encoder function $e$ from the previous subsection can be used in semi-supervised learning. Our framework of semi-supervised learning follows the structure as given in Section 3. Let

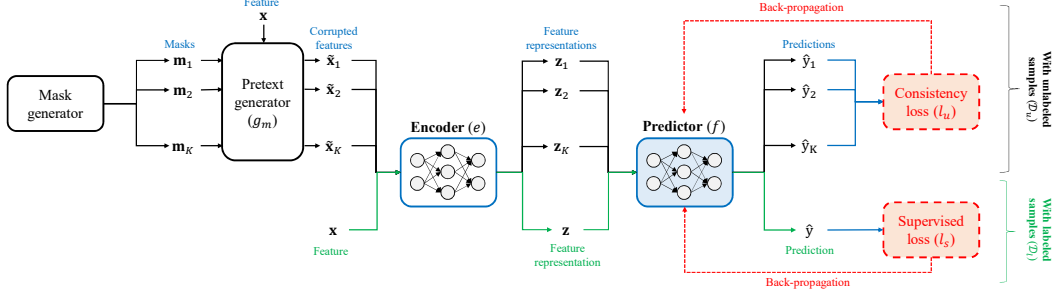

Figure 2: Block diagram of the proposed semi-supervised learning framework on tabular data. For an unlabeled sample $\mathbf{x}$ in $\mathcal{D}_u$, (1) Mask generator generates $K$-number of mask vectors and combine each of them with $\mathbf{x}$ to generate the corrupted samples $\tilde{\mathbf{x}}_k$, $k = 1, ..., K$ via pretext generator ($g_m$), (2) Encoder ($e$) transforms these corrupted samples into latent representations $\mathbf{z}_k$, $k = 1, ..., K$ as $K$ different augmented samples, (3) Predictive model is trained by minimizing the supervised loss on $(\mathbf{x}, y)$ in $\mathcal{D}_l$ and the consistency loss on the augmented samples ($\mathbf{z}_k$, $k = 1, ..., K$) jointly. The block diagram of the proposed self- and semi-supervised learning frameworks on exemplary tabular data can be found in the Supplementary Materials (Figure 2).

$f_e = f \circ e$ and $\hat{y} = f_e(\mathbf{x})$. We train the predictive model $f$ by minimizing the objective function,

$$\mathcal{L}_{final} = \mathcal{L}_s + \beta \cdot \mathcal{L}_u. \tag{7}$$

The supervised loss $\mathcal{L}_s$ is given by

$$\mathcal{L}_s = \mathbb{E}_{(\mathbf{x},y) \sim p_{XY}} \left[ l_s \big( y, f_e(\mathbf{x}) \big) \right], \tag{8}$$

where $l_s$ is the standard supervised loss function, e.g. mean squared error for regression or categorical cross-entropy for classification. The unsupervised (consistency) loss $\mathcal{L}_u$ is defined between original samples ($\mathbf{x}$) and their reconstructions from corrupted and masked samples ($\tilde{\mathbf{x}}$),

$$\mathcal{L}_u = \mathbb{E}_{\mathbf{x} \sim p_X, \mathbf{m} \sim p_{\mathbf{m}}, \tilde{\mathbf{x}} \sim g_m(\mathbf{x},\mathbf{m})} \left[ \big( f_e(\tilde{\mathbf{x}}) - f_e(\mathbf{x}) \big)^2 \right]. \tag{9}$$

Our consistency loss is inspired by the idea in consistency regularizer: encouraging the predictive model $f$ to return the similar output distribution when its inputs are perturbed. However, the perturbation in our framework is learned through our self-supervised framework while in the previous works, the perturbation is from a manually chosen distribution, such as rotation.

For a fixed sample $\mathbf{x}$, the inner expectation in Equation (9) is taken with respect to $p_{\mathbf{m}}$ and $g_m(\mathbf{x}, \mathbf{m})$ and could be interpreted as the variance of the predictions of corrupted and masked samples. $\beta$ is another hyper-parameter to adjust the supervised loss $\mathcal{L}_s$ and the consistency loss $\mathcal{L}_u$. In each iteration of training, for each sample $\mathbf{x} \in \mathcal{D}_u$ in the batch, we create $K$ augmented samples $\tilde{\mathbf{x}}_1, ..., \tilde{\mathbf{x}}_K$ by repeating the operation in Equation (3) $K$ times. Every time the sample $\mathbf{x} \in \mathcal{D}_u$ is used in a batch, we recreate these augmented samples. The stochastic approximation of $\mathcal{L}_u$ is given as

$$\hat{\mathcal{L}}_u = \frac{1}{N_b K} \sum_{i=1}^{N_b} \sum_{k=1}^{K} \left[ \big( f_e(\tilde{\mathbf{x}}_{i,k}) - f_e(\mathbf{x}_i) \big)^2 \right] = \frac{1}{N_b K} \sum_{i=1}^{N_b} \sum_{k=1}^{K} \left[ \big( f(\mathbf{z}_{i,k}) - f(\mathbf{z}_i) \big)^2 \right] \tag{10}$$

where $N_b$ is the batch size. During training, the predictive model $f$ is regularized to make similar predictions on $\mathbf{z}_i$ and $\mathbf{z}_{i,k}$, $k = 1, ..., K$. After training $f$, the output for a new test sample $\mathbf{x}^t$ is given by $\hat{y} = f_e(\mathbf{x}^t)$. Figure 2 illustrates the entire procedure of the proposed semi-supervised framework on tabular data with a pre-trained encoder.

## 5 Experiments

In this section, we conduct a series of experiments to demonstrate the efficacy of our framework (VIME) on several tabular datasets from different application domains, including genomics and clinical data. We use Min-max scaler to normalize the data between 0 and 1. For self-supervised

learning, we compare VIME against two benchmarks, Denoising auto-encoder (DAE) [21] and Context Encoder [22]. For semi-supervised learning, we use the data augmentation method MixUp [10] as the main benchmark. We exclude self- and semi-supervised learning benchmarks that are applicable only to image or language data. As a baseline, we also include supervised learning benchmarks. Additional results with more baselines can be found in the Supplementary Materials. In the experiments, self- and semi-supervised learning methods use both labeled data and unlabeled data, while the supervised learning methods only use the labeled data. Implementation details and sensitivity analyses on three hyperparameters $(p_m, \alpha, \beta)$ can be found in the Supplementary Materials (Section 5 & 6). The implementation of VIME can be found at `https://bitbucket.org/mvdschaar/mlforhealthlabpub/src/master/alg/vime/` and at `https://github.com/jsyoon0823/VIME`.

## 5.1 Genomics data: Genome-wide polygenic scoring

In this subsection, we evaluate the methods on a large genomics dataset from UK Biobank consisting of around 400,000 individuals' genomics information (SNPs) and 6 corresponding blood cell traits: (1) Mean Reticulocyte Volume (MRV), (2) Mean Platelet Volume (MPV), (3) Mean Cell Hemoglobin (MCH), (4) Reticulocyte Fraction of Red Cells (RET), (5) Plateletcrit (PCT), and (6) Monocyte Percentage of White Cells (MONO). The features of the dataset consist of around 700 SNPs (after the standard p-values filtering process), where each SNP, taking value in $\{0, 1, 2\}$, is treated as a categorical variable (with three categories). Here, we have 6 different blood cell traits to predict, and we treat each of them as an independent prediction task (selected SNPs are different across different blood cell traits). Detailed data descriptions are provided in the Supplementary Materials (Section 2). Note that all the variables are categorical features.

To test the effectiveness of self- and semi-supervised learning in the small labeled data setting, VIME and benchmarks are tasked to predict the 6 blood cell traits while we gradually increase the number of labeled data points from 1,000 to 100,000 samples while using the remaining data as unlabeled data (more than 300,000 samples). We use a linear model (Elastic Net [34]) as the predictive model due to their superior performance in comparison to other non-linear models such as multi-layer perceptron and random forests [35] on genomics datasets.

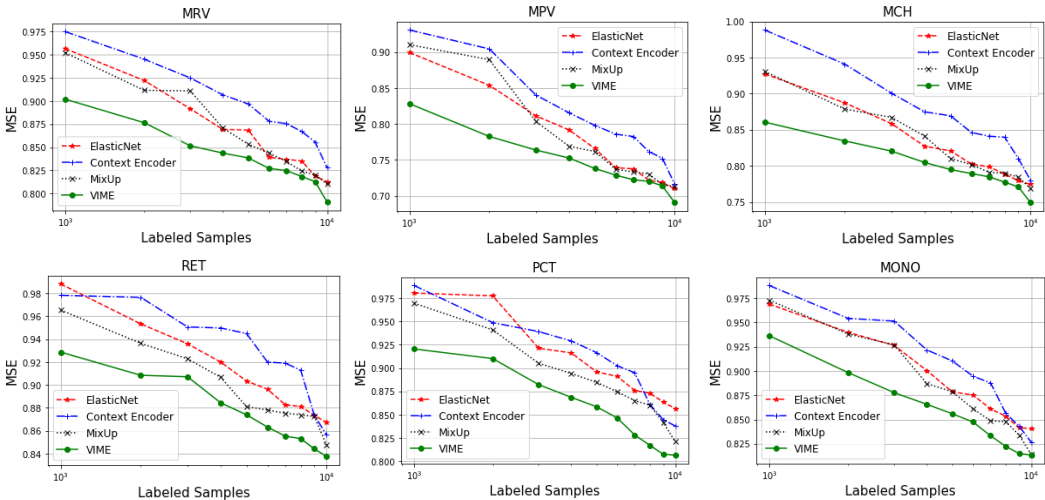

Figure 3: MSE performances on 6 different blood cell traits across different sizes of the labeled genomics dataset (lower the better). Note that x-axis is a log-scale.

In Figure 3, we show the MSE performance (y-axis) against the number of labeled data points (x-axis, in log scale) increasing from 1,000 to 10,000[3]. The proposed model (VIME) outperforms all the benchmarks, including purely supervised method ElasticNet, the self-supervised method Context Encoder and the semi-supervised method MixUp. In fact, in many cases VIME shows similar

performances to the benchmarks even when it has access to only half as many labeled data points (as the benchmarks).

## 5.2  Clinical data: Patient treatment prediction

In this subsection, we evaluate the methods on clinical data, using the UK and US prostate cancer datasets (from Prostate Cancer UK and SEER datasets, respectively). The features consist of patients' clinical information (e.g. age, grade, stage, Gleason scores) - total 28 features. We predict 2 possible treatments of UK prostate cancer patients (1) Hormone therapy (whether the patients got hormone therapy), (2) Radical therapy (whether the patient got radical therapy). Both tasks are binary classification. In the UK prostate cancer dataset, we only have around 10,000 labeled patients samples. The US prostate cancer dataset contains more than 200,000 unlabeled patients samples, twenty times bigger than the labeled UK dataset. We use 50% of the UK dataset (as the labeled data) and the entire US dataset (as the unlabeled data) for training, with the remainder of the UK data being used as the testing set. We also test three popular supervised learning models: Logistic Regression, a 2-layer Multi-layer Perceptron and XGBoost.

Table 1 shows that VIME results in the best prediction performance, outperforming the benchmarks. More importantly, VIME is the only self- or semi-supervised learning framework that significantly outperforms supervised learning models. These results shed light on the unique advantage of using VIME in leveraging a large unlabeled tabular dataset (e.g. the US dataset) to strengthen a model's predictive power. Here we also demonstrate that VIME can perform well even when there exists a distribution shift between the UK labeled data and the US unlabeled data (see the Supplementary Materials (Section 2) for further details).

Table 1: AUROC Performances of patient treatment predictions on Hormone and Radical therapy (higher the better). (Mean $\pm$ Standard deviations are computed over 10 runs)

| Type | Models | Hormone | Radical |
|---|---|---|---|
| SL | Logistic Regression | .8371$\pm$.0013 | .8036$\pm$.0015 |
| | 2-layer Perceptron | .8351$\pm$.0023 | .8146$\pm$.0022 |
| | XGBoost | .8423$\pm$.0018 | .8166$\pm$.0011 |
| Self-SL | DAE | .8335$\pm$.0049 | .8144$\pm$.0061 |
| | Context Encoder | .8308$\pm$.0051 | .8134$\pm$.0066 |
| Semi-SL | MixUp | .8448$\pm$.0021 | .8214$\pm$.0029 |
| | VIME | **.8602$\pm$.0029** | **.8391$\pm$.0021** |

## 5.3  Public tabular data

To further verify the generalizability and allow for reproducibility of our results, we compare VIME with the benchmarks using three public tabular datasets: MNIST (interpreted as a tabular data with 784 features), UCI Income and UCI Blog. We use 10% of the data as labeled data, and the 90% of the remaining data as unlabeled data. Prediction accuracy on a separate testing set is used as the metric for all three datasets. As shown in Table 2 (Type - Supervised models, Self-supervised models, Semi-supervised models and VIME), VIME achieves the best accuracy regardless of the application domains. These results further confirm the superiority of VIME in a diverse range of tabular datasets.

## 5.4  Ablation study

In this section, we conduct an ablation study to analyze the performance gain of each component in VIME on the tabular datasets introduced in Section 5.3. We define three variants of VIME:

• **Supervised only:** Exclude both self- and semi-supervised learning parts (i.e. 2-layer perceptron)
• **Semi-SL only:** Exclude self-supervised learning part (i.e. remove the encoder in Figure 2)
• **Self-SL only:** Exclude semi-supervised learning part (i.e. $\beta = 0$). More specifically, we first train the encoder via self-supervised learning. Then, we train the predictive model with loss function (in Equation (7) with $\beta = 0$ (only utilizing the labeled data).

Table 2: Prediction accuracy of the methods on UCI Income, MNIST and UCI Blog datasets (Mean $\pm$ Std are computed over 10 runs).

| Type | Models | Income | MNIST | Blog |
|---|---|---|---|---|
| **Supervised models** | Logistic Regression | .8425±.0013 | .8989±.0023 | .6915±.0029 |
| | 2-layer Perceptron | .8520±.0023 | .9387±.0014 | .7972±.0058 |
| | XGBoost | .8623±.0021 | .9413±.0026 | .7975±.0030 |
| **Self-supervised models** | DAE | .8578±.0028 | .9431±.0032 | .8001±.0039 |
| | Context Encoder | .8611±.0027 | .9455±.0048 | .8033±.0051 |
| **Semi-supervised models** | MixUp | .8701±.0021 | .9461±.0023 | .8088±.0038 |
| **Variants of VIME** | Supervised only | .8520±.0023 | .9387±.0014 | .7972±.0058 |
| | Self-SL only | .8599±.0026 | .9406±.0019 | .8147±.0037 |
| | Semi-SL only | .8771±.0031 | .9548±.0023 | .8361±.0041 |
| **VIME** | | **.8804±.0030** | **.9577±.0022** | **.8389±.0044** |

Table 2 (Type - Variants of VIME and VIME) shows that both **Self-SL only** and **Semi-SL only** show performance gains compared with **Supervised only**, and VIME is always better than its variants. Every component in VIME can improve the performance of a predictive model, and the best performance is achieved when they work collaboratively in our unified framework. We note that **Self-SL only** leads to a larger performance drop than **Semi-SL only** because in the former the predictive model is trained solely on a small labeled dataset without the unsupervised loss function $\mathcal{L}_u$, while in the latter the predictive model is trained via minimizing both losses but without the encoder. Additional ablation study can be found in the Supplementary Materials.

# 6 Discussions: Why does the proposed model (VIME) need for tabular data?

Image and tabular data are very different. The spatial correlations between pixels in images or the sequential correlations between words in text data are well-known and consistent across different datasets. By contrast, the correlation structure among features in tabular data is unknown and varies across different datasets. In other words, there is no "common" correlation structure in tabular data (unlike in image and text data). This makes the self- and semi-supervised learning in tabular data more challenging. Note that promising methods for image domain do not guarantee the favorable results on tabular domain (vice versa). Also, most augmentations and pretext tasks used in image data are not applicable to tabular data; because they directly utilize the spatial relationship of the image for augmentation (e.g., rotation) and pretext tasks (e.g., jigsaw puzzle and colorization). To transfer the successes of self- and semi-supervised learning from image to tabular domains, proposing applicable and proper pretext tasks and augmentations for tabular data (our main novelty) is critical. Note that better augmentations and pretext tasks can *significantly* improve self- and semi-supervised learning performances.

## Broader Impact

Tabular data is the most common data type in the real-world. Most databases include tabular data such as demographic information in medical and finance datasets and SNPs in genomic datasets. However, the tremendous successes in deep learning (especially in image and language domains) has not yet been fully extended to the tabular domain. Still, in the tabular domain, ensembles of decision trees achieve the state-of-the-art performance. If we can efficiently extend the successful deep learning methodologies from images and language to tabular data, the application of machine learning in the real-world can be greatly extended. This paper takes a step in this direction for self- and semi-supervised learning frameworks which recently have achieved significant successes in images and language. In addition, the proposed tabular data augmentation and representation learning methodologies can be utilized in various fields such as tabular data encoding, balancing the labels of tabular data, and missing data imputation.

## Acknowledgements and Funding Sources

The authors would like to thank the reviewers for their helpful comments. This work was supported by the National Science Foundation (NSF grant 1722516), the US Office of Naval Research (ONR), and GlaxoSmithKline (GSK).

## Footnotes

[1]Tabular data is a database that is structured in a tabular form. It arranges data elements in vertical columns (features) and horizontal rows (samples).

[2]Subscript $j$ represents the $j$-th element of the vector.

[3]The performances for 10,000 to 100,000 range can be found in the Supplementary Materials (Section 3)

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
