[Supplementary Material]

# Supplementary Materials -
# VIME: Extending the Success of Self- and Semi-supervised Learning to Tabular Domain

**Jinsung Yoon**
Google Cloud AI, UCLA
jinsungyoon@google.com

**Yao Zhang**
University of Cambridge
yz555@cam.ac.uk

**James Jordon**
University of Oxford
james.jordon@wolfson.ox.ac.uk

**Mihaela van der Schaar**
University of Cambridge
UCLA, Alan Turing Institute
mv472@cam.ac.uk

## 1 Additional block diagrams

Figure 1: The systematic self- and semi-supervised learning frameworks in VIME. Self-supervised learning trains an encoder to extract informative representations on the unlabeled data. Semi-supervised learning uses the trained encoder in learning a predictive model on both labeled and unlabeled data.

(a) Self-supervised learning          (b) Semi-supervised learning
                                          (on unlabeled data)

Figure 2: The proposed self- and semi-supervised learning frameworks on exemplary tabular data. (a) In self-supervised learning, the mask and feature vector estimators try to recover the original unlabeled feature $[1, 7, 4, 3, 5, 1]$ and the mask vector $[0, 0, 1, 0, 1, 0]$ from its masked and corrupted variant $[1, 7, 6, 3, 2, 1]$. (b) In semi-supervised learning, the encoder transforms some masked and corrupted variants into some augmented representations for training a consistent predictive model.

Figure 3: The proposed data corruption procedure. Original feature matrix ($\mathbf{X}$) consists of four samples $\mathbf{x}_i, i = 1..., 4$, where each row/column represents a sample/feature, and the features in each sample are represented by the same color. (1) We randomly shuffle the original data within each feature to construct randomly shuffled feature matrix ($\bar{\mathbf{X}}$) (i.e. column-wise shuffling). (2) We randomly sample the mask matrix ($\mathbf{M}$) from a Bernoulli distribution and compute its counterpart $(\mathbf{1} - \mathbf{M})$. (3) We construct the corrupted feature matrix ($\tilde{\mathbf{X}}$) by the following equation: $\tilde{\mathbf{X}} = \mathbf{M} \odot \bar{\mathbf{X}} + (\mathbf{1} - \mathbf{M}) \odot \mathbf{X}$ where each row of $\tilde{\mathbf{X}}$ is given as $\tilde{\mathbf{x}}_i = \mathbf{m}_i \odot \bar{\mathbf{x}}_i + (\mathbf{1} - \mathbf{m}_i) \odot \mathbf{x}_i$, and $\odot$ is element-wise multiplication.

## 2 Detailed data descriptions

In the experiment section of the main manuscript, we evaluate VIME and its benchmarks on 11 datasets (6 genomics, 2 clinical, and 3 public datasets). Here, we provide the basic data statistics for the 11 used datasets in Table 1.

Table 1: Basic data statistics (the number of samples and dimensions) for 11 tabular datasets (6 genomics, 2 clinical, and 3 public datasets). *Dims* represents dimensions.

| Genomics | | | Clinical | | | Public | | |
|---|---|---|---|---|---|---|---|---|
| Labels | No | *Dims* | Locations | No | *Dims* | Datasets | No | *Dims* |
| MRV | 3932,45 | 682 | UK | 10,086 | 28 | Income | 48,842 | 108 |
| MPV | 391,598 | 691 | US | 240,486 | 27 | MNIST | 70,000 | 784 |
| MCH | 406,517 | 700 | | | | Blog | 60,021 | 280 |
| RET | 396,720 | 604 | | | | | | |
| PCT | 390,803 | 743 | | | | | | |
| MONO | 403,994 | 681 | | | | | | |

### 2.1 Genomics data

The UK Biobank genomic datasets consist of around 500,000 individuals' genomic information along with 6 different blood cell traits:

- Mean Reticulocyte Volume (MRV),
- Mean Platelet Volume (MPV),
- Mean Cell Hemoglobin (MCH),
- Reticulocyte Fraction of Red Cells (RET),
- Plateletcrit (PCT),
- Monocyte Percentage of White Cells (MONO).

To select the significantly associated variants (SNPs) for each blood cell trait among the entire genetic variants, we take the *lead* SNPs (i.e. minimum p-value SNP) from each independent locus that is discovered at genome-wide significance (i.e. which has at least one SNP with p-values $< 8.31e\text{-}9$). The selected SNPs and the corresponding blood cell trait together form an independent labeled dataset. We note that the selected SNPs vary over the blood cell traits since each trait is associated with different genetic variants. For each genomic dataset, we use 10% of the samples as the testing data on the downstream task. Among the remaining 90% of samples, we use the given number of samples (from 1,000 to 100,000) as the labeled dataset and the remaining samples as the unlabeled data (more than 300,000 samples).

### 2.2 Public data

The details of three public datasets can be found in the following hyperlinks: (1) UCI Income, (2) MNIST, (3) UCI Blog. All three datasets are given as with separate training and testing sets. We use 10% of the training set as the labeled data and the remaining data as the unlabeled data. The testing set is used to evaluate the trained models in the downstream tasks.

### 2.3 Clinical data

In this paper, we use two clinical datasets from two different countries: (1) United Kingdom (UK), (2) United States (US). UK dataset comes from *Prostate Cancer UK* data and the US dataset comes from *SEER* data (details can be found in the hyperlink). In the UK prostate cancer dataset, we have 28 features, including age, Prostate-Specific Antigen (PSA), clinical stages, primary and secondary Gleason scores, grade, and treatment information (radical and hormone therapy). For the US prostate cancer dataset, we extract all the features that exist in the UK dataset except for the treatment information (not exist in US prostate cancer dataset). The treatment information becomes the label

for the predictive model. We use 50% of the UK prostate cancer dataset as the testing data for the downstream task (two treatment estimations), and the remaining 50% of the UK prostate cancer data as the labeled data. The entire US prostate cancer dataset is used as the unlabeled data. Table 2 describes the feature distributions of UK and US prostate cancer datasets. We can easily identify the distribution mismatch between UK and US datasets.

Table 2: Basic data statistics of two clinical datasets: (1) UK prostate cancer data, (2) US prostate cancer data. For continuous variables, we report the median value with 25% and 75% percentiles. For categorical variables, we report the number of patients and its ratio.

| Variables | | UK data | US data |
|---|---|---|---|
| Age | | 70 (64 - 76) | 66 (60 - 73) |
| PSA | | 12 (8 - 21) | 7 (5 - 11) |
| Clinical Stage | 1 | 5419 (53.7%) | 3900 (1.6%) |
| | 2 | 3212 (31.8%) | 198245 (82.4%) |
| | 3 | 1378 (13.7%) | 18834 (7.8%) |
| | 4 | 77 (0.8%) | 19507 (8.1%) |
| Primary Gleason | 1 | 41 (0.4%) | 1729 (0.7%) |
| | 2 | 340 (3.4%) | 8361 (3.5%) |
| | 3 | 6116 (60.6%) | 167269 (69.6%) |
| | 4 | 3076 (30.5%) | 55604 (23.1%) |
| | 5 | 513 (5.1%) | 7523 (3.1%) |
| Secondary Gleason | 1 | 43 (0.4%) | 546 (0.2%) |
| | 2 | 321 (3.2%) | 9158 (3.8%) |
| | 3 | 4562 (45.2%) | 129417 (53.8%) |
| | 4 | 4097 (40.6%) | 84791 (35.3%) |
| | 5 | 1063 (10.5%) | 16574 (6.9%) |
| Total Gleason Score | $\leq 5$ | 474 (4.7%) | 12389 (5.2%) |
| | 6 | 2915 (28.9%) | 99100 (41.2%) |
| | 7 | 4470 (44.3%) | 90397 (37.6%) |
| | 8 | 1004 (10.0%) | 20142 (8.4%) |
| | 9 | 1107 (11.0%) | 16144 (6.7%) |
| | 10 | 116 (12%) | 2344 (1.0%) |
| Grade | 1 | 3362 (33.3%) | 13808 (5.7%) |
| | 2 | 2983 (29.6%) | 99622 (41.4%) |
| | 3 | 1513 (15.0%) | 126505 (52.6%) |
| | 4 | 1005 (10.0%) | 551 (0.2%) |
| | 5 | 1223 (12.1%) | 0 (0.0%) |
| Therapy | Hormone | 3176 (31.5%) | N/A |
| | Radical | 4914 (48.8%) | N/A |
| | Others | 1996 (19.8%) | N/A |

# 3 Additional experiments

## 3.1 Performances on 6 genomics datasets with different numbers of labeled samples (from 1,000 to 100,000)

Figure 4: MSE performances on 6 different blood cell traits across different sizes of the labeled genomics datasets (lower the better). The range of the x-axis is given as [1,000, 100,000] in the log scale.

## 3.2 Performances on MNIST dataset with different numbers of labeled samples (from 100 to 5,000)

Figure 5: Accuracy performances on MNIST dataset across different sizes of the labeled samples (higher the better). The range of the x-axis is given as [100, 1,000] (left) and [1,000, 5,000] (right) in the log scale.

## 3.3 Qualitative performance analyses of pretext models on UCI Income and MNIST datasets

| Original data | Corrupted data | Mask vector | Recovered data | Estimated mask vector |

Figure 6: Qualitative pretext model performances in the pretext tasks, feature and mask vector estimations, on the UCI Income dataset. (first column: original data, second column: corrupted data, third column: mask vector, fourth column: recovered data, fifth column: estimated mask vector). Both the feature and mask vectors are recovered well. 108 feature vectors are visualized as $12 \times 9$ matrices.

Proposed model (VIME)             Semi-SL Only (without encoder)

Figure 7: t-SNE analyses on the augmented samples in VIME (left) and **Semi-SL Only** variant (right) on MNIST dataset. The t-SNE embeddings in VIME are much more separable than the ones in **Semi-SL Only** variant which qualitatively demonstrates the necessity of Self-SL in VIME.

## 3.4 Performances of pretext tasks

In this paper, we introduce two pretext tasks in the proposed self-supervised learning framework: (1) feature vector estimation, and (2) mask vector estimation. In this subsection, we quantitatively and qualitatively assess the performances of the pretext models (encoder, feature and mask estimators) in the pretext tasks. We use UCI Income dataset as an example. In the pretext distribution, we introduce 20% masking and corruptions ($p_m = 0.2$) to the original data.

Original data matrix    Corrupted data matrix    Mask matrix    Recovered data matrix    Estimated mask matrix

Figure 8: Qualitative analysis on the pretext tasks, feature and mask vector estimations, on UCI Income dataset. (first column: original data matrix, second column: corrupted and masked data matrix, third column: mask matrix, fourth column: recovered data matrix, fifth column: estimated mask matrix). Each row/column represents the sample/feature.

As can be seen in Figure 8, the recovered data matrix (fourth column) is quite similar to the original data matrix (first column). And the estimated mask matrix (fifth column) is also reasonably similar to the original mask matrix (third column). Table 3 reconfirms quantitatively that the pretext models recover the feature and mask vectors successfully in 4 different datasets.

Table 3: Performances of the pretext tasks, feature and mask vector estimation, for 4 tabular datasets. AUROC (higher the better) / MSE (lower the better) is used as a metric for mask/feature estimation pretext task.

| Pretext tasks (Metrics) | Mask estimation (AUROC) | Feature estimation (MSE) |
|---|---|---|
| Income | $.9639 \pm .0009$ | $.0093 \pm .0008$ |
| MNIST | $.9208 \pm .0006$ | $.0082 \pm .0006$ |
| Blog | $.9354 \pm .0002$ | $.0087 \pm .0003$ |
| Clinical | $.9485 \pm .0025$ | $.0156 \pm .0004$ |

Since the pretext models solve the two challenging pretext tasks, feature and mask vector estimations, the learned representations in VIME are more informative than the ones in the self-supervised learning benchmarks. As a comparison, Context Encoder, taking the mask vector as an input, can easily identify the corrupted features. Also, in the case of Context Encoder, only the reconstruction error for the corrupted components is used to train the networks, and the resulting representations may fail to preserve the information in the non-corrupted features. In DAE, the Gaussian noise is easy to identify when it is added to the categorical features often found in tabular data.

# 4 The limitations of MixUp

MixUp [1] is a data augmentation method where an augmented sample is generated by taking a convex combination of two original samples. MixUp may work well if the original data manifold is convex. However, if the original data manifold is non-convex, it is highly likely that the augmented data generated by MixUp is out of the support of the original data distribution (see Fig. 9 (a)). The manifold of the original tabular data is often non-convex (due to the categorical variables); thus, MixUp is not an effective data augmentation method for tabular data.

**Medical Example:** MixUp can generate the augmented samples which lie outside the support of the original tabular data distribution. Let us assume that we have two variables: (1) age (continuous), (2) prostate cancer (binary). Also, let us assume that the senior citizens above 70 years old are the only population group that has prostate cancer. As can be seen in Fig. 9 (b), in this case, the augmented

Figure 9: The failure scenarios of MixUp as a data augmentation method for tabular data: (a) Non-convex data manifold, (b) Mixed-type variables. Note that the augmented data points ($\times$) are located out of the original data manifold in both cases.

data points generated by MixUp (using convex combinations) are prostate cancer patients below 70 years old.

## 5 Implementation details

VIME consists of 4 separate networks: (1) encoder, (2) mask vector estimator, (3) feature vector estimator, (4) predictive model. We use a multi-layer perceptron (MLP) as the baseline architecture for all the networks. The number of hidden units, the number of layers and hyperparameters in VIME and benchmark methods (including standard supervised models such as XGBoost and ElasticNet) are optimized by cross-validation, and we choose the ones with the best downstream task performance on validation data (10% of the training data). The number of hidden units is selected among $\{d/3, d/2, d, 2d, 3d\}$, and the number of layers is selected among $\{1, 2, 3, 4, 5\}$. We use ReLU as the activation function at each hidden layer. Also, we select $K$, the number of augmented samples, among $\{2, 3, 5, 10, 15, 20\}$. The selection of the four hyperparameters $(p_m, \alpha, \beta, K)$ in VIME are discussed in a separate section (Section 6).

VIME uses 4 different losses to train the networks: (1) mask vector estimation loss $l_m$, (2) reconstruction loss $l_r$, (3) supervised loss $l_s$, and (4) unsupervised loss $l_u$. For $l_m$ and $l_r$, we use binary-cross entropy and mean squared errors as the loss function respectively, regardless of the problems (whether it is a classification or regression problem). For $l_s$, we use cross-entropy and mean squared errors for classification and regression problems, respectively. For $l_u$, we use the mean squared errors on the output predictions for regression and logits for classification problems.

## 6 Hyperparameter analyses

VIME has four important hyperparameters: (1) $p_m$ - adjusts the proportion of masked and corrupted features, (2) $\alpha$ - adjusts the trade-off between mask estimation loss and reconstruction loss, (3) $\beta$ - adjusts the trade-off between supervised loss and unsupervised (consistency) loss, (4) $K$ - adjusts the number of augmented samples. In Fig. 10, we vary four hyperparameters $(p_m, \alpha, \beta, K)$ and report the sensitivity analyses on MNIST dataset. We vary $p_m \in [0.1, 0.9]$, $\alpha \in [0.1, 10]$, $\beta \in [0.1, 10]$, and $K \in \{2, 3, 5, 10, 15, 20\}$.

As can be seen in Fig. 10, with a large $p_m$, the combinations of the encoder, mask vector estimator, and feature vector estimator may fail to solve the pretext tasks. On the other hand, if $p_m$ is too small, the three networks can solve the pretext easily, and the resulting representations may not be that informative. $\alpha$ adjusts the balance between two pretext tasks. With an inappropriate $\alpha$, the proposed self-supervised learning model only focuses on mask vector estimation or feature vector estimation; thus, the synergy of those two pretext tasks may decrease. $\beta$ adjusts the balance between self- and semi-supervised losses. With too large or too small $\beta$, the proposed semi-supervised learning model only focuses on the loss on the labeled or unlabeled data; thus, the synergy of those two datasets may

Figure 10: Accuracy performances on MNIST dataset across different values of the hyperparameters (a) $p_m$, (b) $\alpha$, (c) $\beta$, (d) $K$. The ranges of the x-axis for $\alpha$ and $\beta$ are given as $[0.1, 10]$ in the log scale.

drop. $K$ adjusts the number of augmented unlabeled samples. With too large $K$, the computational complexity would significantly increase. On the other hand, with too small $K$, the unsupervised loss ($l_u$) would be noisy. In our experiments, we choose $p_m \in [0.1, 0.9]$, $\alpha \in [0.1, 10]$, $\beta \in [0.1, 10]$, and $K \in \{2, 3, 5, 10, 15, 20\}$ using cross-validation.

## 7 Additional discussions

### 7.1 Why is the proposed model (VIME) better?

In this section, we further analyze why our VIME is superior to the various benchmarks. First, compared with supervised learning, VIME benefits from its representation learning by training a model on not only a small labeled dataset but also a large unlabeled dataset.

In self-supervised learning, the learned representations in VIME are more informative than the benchmarks, since the encoder solves more challenging pretext tasks, feature and mask vector estimations. As a comparison, Context Encoder [2], taking the mask vector an input, can easily identify those corrupted features. In Denoising auto-encoder [3], the Gaussian noise is easy to identify when it is added to tabular data with some categorical features.

In semi-supervised learning, the data augmentation scheme in VIME generates augmented samples on the manifold of the original tabular data even if the manifold is non-convex. In contrast, the augmented samples in MixUp [1], generated by taking a convex combination of two original samples, may lie outside a non-convex data manifold.

### 7.2 Why the encoder is needed?

The importance of the encoder is demonstrated by comparing the performances of VIME and **Semi-SL only**, the variant of VIME without the encoder. Using t-SNE analyses [4] on the UCI Income dataset, we visually compare the distribution of augmented samples with and without the encoder. We introduce 20% corruption ($p_m = 0.2$) to the original data via the pretext generator ($g_m$) for data augmentation.

| Proposed model (VIME) | Semi-SL Only (without encoder) |

Figure 11: t-SNE analyses on the augmented samples by VIME (left) and Semi-SL Only variant (right).

As can be seen in Figure 11, the augmented samples from the encoder in VIME are much more separable and representative of the labels than the ones in the **Semi-SL Only** variant. This is due to the way $e$ is trained, the representations, $\mathbf{z}$, generated by $e$ contain discriminative information about the mask vector and information about how to impute the masked and corrupted features. A predictive model built on $\mathbf{z}$ has a more informative input and hence requires a less complex model to minimize the given losses in our semi-supervised learning framework.

# 8  Additional ablation study

To further highlight the utility of the novel mask prediction task, we demonstrate the additional ablation study. As can be seen in Table 4, without mask vector estimation, the performance improvements by self-supervised learning degraded 21.4%, 60.4%, 14.8% for Income, MNIST, and Blog datasets, respectively.

Table 4: Additional ablation study on UCI Income, MNIST and UCI Blog datasets. (Mean $\pm$ Std are computed over 10 runs)

| Models | Income | MNIST | Blog |
|---|---|---|---|
| Supervised only | .8520±.0023 | .9387±.0014 | .7972±.0058 |
| Self-SL only | .8599±.0026 | .9406±.0019 | .8147±.0037 |
| Self-SL only w/o Mask estimates | .8582±.0031 | .9394±.0018 | .8121±.0040 |
| Self-SL only w/o Feature estimates | .8433±.0033 | .9301±.0022 | .7919±.0067 |
| Semi-SL only | .8771±.0031 | .9548±.0023 | .8361±.0041 |
| VIME with Gaussian noise augmentation | .8627±.0035 | .9481±.0041 | .8253±.0044 |
| **VIME** | **.8804±.0030** | **.9577±.0022** | **.8389±.0044** |

Directly estimating the values of the corrupted samples would be a reasonable but hard pretext task (imputation problem). On the other hand, finding where the missing data is (binary prediction problem) would be an easier pretext task that can help learning better representations jointly with the harder (but relevant) pretext task (imputation). With the collaborations of easy and hard pretext tasks, the self-supervised learning framework can learn better representations.

Also, We perform an additional experiment in which we add Gaussian noise to the original data and treat this as the augmented samples (VIME with Gaussian noise augmentation). As can be seen in Table 4, performances with Gaussian noise augmentations are consistently worse than the performance of the original VIME.

# 9 Additional baselines

In this section, we includes four additional self-supervised learning baselines (Split-brain [5], SimCLR [6], TabNet [7] and TaBERT [8]) that can be extended to tabular domain. As can be seen in Table 5, the performance of split-brain on MNIST data is 0.9411 which is worse than DAE (0.9431) and Context Encoder (0.9455). Also, the performance of SimCLR on Blog data is 0.8044 which is worse than the performance of the VIME variant Self-SL Only (0.8147).

We acknowledge that self/semi-supervised learning is well-studied in the image and language domains. However, as can be seen in Table 5, the state-of-the-art self/semi-supervised learning models for image and language domains underperform VIME in the tabular setting. This demonstrates that new self/semi-supervised learning models are necessary for tabular domain.

Table 5: Prediction accuracy of the methods on UCI Income, MNIST and UCI Blog datasets with additional baselines. (Mean $\pm$ Std are computed over 10 runs)

| Type | Models | Income | MNIST | Blog |
|---|---|---|---|---|
| **Self-supervised models** | DAE | .8578±.0028 | .9431±.0032 | .8001±.0039 |
| | Context Encoder | .8611±.0027 | .9455±.0048 | .8033±.0051 |
| | Split-Brain [5] | .8591±.0032 | .9411±.0039 | .8037±.0048 |
| | SimCLR [6] | .8637±.0025 | .9437±.0041 | .8044±.0043 |
| | TabNet [7] | .8637±.0033 | .9487±.0039 | .8114±.0044 |
| | TaBERT [8] | .8653±.0037 | .9463±.0051 | .8127±.0041 |
| **Variants of VIME** | Self-SL only | .8599±.0026 | .9406±.0019 | .8147±.0037 |
| **VIME** | | **.8804±.0030** | **.9577±.0022** | **.8389±.0044** |