[Reviews · NeurIPS 2020]

Review 1

Summary and Contributions: This work proposes a self-supervised framework for representation learning with tabular data. The authors propose a multi-headed self-supervised training model that first corrupts (augments) the input tabular data using a binary mask, and then one head reconstructs the mask while the other head reconstructs the uncorrupted data. In addition, the authors use a standard supervised loss function for data that contain labels, an addition that makes this model applicable to semi-supervised learning. The authors demonstrate the effectiveness of their multi-headed reconstruction pretext task on a genomics dataset, patient treatment dataset, and two tabular benchmark datasets (UCI Income & Blog) as well as MNIST treated as tabular data.

Strengths: Overall, machine learning on tabular data is an understudied problem, and this paper lays out a clear and justifiable explanation for the development of their self-supervised pretraining approach for tabular data. The paper proposes a novel 2-part reconstruction task for masked tabular data: where both reconstructing the mask itself and the unmasked input data are the two feedback mechanisms for the self-supervised learning. The paper studies a unique set of genomics and patient treatment datasets that tie in nicely with the original motivation of the paper. The experimental results look promising, and the authors include a few ablations to better understand the benefit of the semi-supervised learning component. The applicability tabular data and the empirical evaluations are the primary strengths of this work.

Weaknesses: My central concern for this paper is the misalignment between the motivation and methodology. As motivation, the authors argue that self-supervised CV and **NLP** “algorithms are not effective for tabular data.” The proposed model, though, is effectively the binary masked language model whose variants pervade self-supervised NLP research (e.g. WordNet, BERT, etc). Granted, instead of masking words, the proposed models are masking tabular values, but this is performing a very similar pretext task. In fact, there is concurrent work that learns tabular representations using a BERT model [1]. At the very least, I think it’s worth a discussion of how this masked entry model is similar to a masked language model. I believe this paper also overlooks [2] as related research. Line 167-168: The justification for using the two-component pretext tasks is that it is a difficult individual task. Did you explore using only one of the two-components? Line 195: Is it true that the correlation structure is less obvious in tabular data than in images or text? The semi-supervised learning aspect of this paper described in S4.2 (using a weighted combination of an unsupervised loss function and a supervised loss function) is well established, e.g. [3], and I think this paper could focus more on the novelty of the pretext tasks for tabular data. It would be interesting to experiment and measure the performance of alternative corruption (augmentation) models and their impact on different kinds of tabular data. [1] TaBERT: Pretraining for Joint Understanding of Textual and Tabular Data, https://arxiv.org/abs/2005.08314 [2] TabNet https://arxiv.org/abs/1908.07442 [3] Yves Grandvalet and Yoshua Bengio. Semi-supervised learning by entropy minimization. In Advances in Neural Information Processing Systems, 2005.

Correctness: The methods and empirical methodology appear correct.

Clarity: Overall the paper is clearly written. I believe Figure 1 and Figure 2 could be easily combined. Typos: Line 64: “multivie” Line 225: Missing space after sentence.

Relation to Prior Work: (discussed in "Weaknesses")

Reproducibility: Yes

Additional Feedback: UPDATE After reading all reviews and the author's responses: they addressed my primary criticisms of (i) leaving out related works (ii) overclaiming on their novelty (iii) clarity issues. Some of these concerns were also shared with other reviewers. With the proposed updates, I think this paper will be a worthwhile contribution.


Review 2

Summary and Contributions: This manuscript contributes self and semi-supervised approaches well suited to tabular data. The point being that tabular data does not come with obvious invariants and corresponding transformations that can be used to create selecf supervision. The contributed method relies creating representations that facilitate learning.

Strengths: The work contributes a new reconstruction loss for unsupervised training of representations. This loss extends auto-encoders practice with a pretext task that uses the marginal distribution of features. It can then be used to help training intermediate representations in a semi-supervised setting, to improve prediction, adapting existing frameworks The manuscript contributes empirical benchmarks on a genomic dataset as well as clinical data and a few UCI tabular datasets, demonstrating some increasing in performance.

Weaknesses: The manuscript is tackling tabular data, however it avoids the problem of categorical entries, which are frequent in such data. In particular, the squared loss is used (eq 6), which is not very relevant for categorical data. Likewise, in the experimental validation, the data used do not seem to have categorical data, although the UK Biobank does have categorical features, beyond genomics. As a baseline for the genomics experiments, it would have been interesting to use a PCA to learn representations. In genomics, such a simple model often performs well. With regards to the encoder baseline: were the data centered and normed before fitting an auto-encoder? Indeed, in the absence of standardization, the reconstruction loss is brittle.

Correctness: It seems methodologically correct, but the baselines need to be run well, in particular standardization of the data.

Clarity: Overall, the paper is well written. I must confess that it took me a while to understand that the "supervised only" line of table 2 was the same thing as the "2-layer perceptron".

Relation to Prior Work: The relation to other works is well discussed.

Reproducibility: Yes

Additional Feedback:


Review 3

Summary and Contributions: This paper extends the self/semi-supervised learning to the tabular domain. VIME is proposed to estimate the mask as a pretext task. Experiments on related datasets show their superiority. After reading the response, the authors resolve part of my concerns. I revise the overall score to 6.

Strengths: (1) The extension to the tabular domain with mask estimation is interesting and useful. (2) The authors conduct extensive experiments. Experimental results look good compared with Mix-up.

Weaknesses: (1) I think the novelty is limited. As introduced in the paper, self/semi-supervised learning has already been thoroughly investigated in other domains, including the image and language. Tabular domain aside, feature vector estimation is common in auto-encoder, and the novelty of the proposed mask estimation is not good enough. I think the existing Gaussian noise based augmentation and estimation is very similar except the difference in distribution. (2) The motivation that you generate the masked samples by Eq.(3) is unclear. Why you add the first term, especially after the shuffle operation? (3) I think the mask m does not need to be binary. Have you ever tried other distributions, such as the Gaussian distribution? You only claim that your approach is more difficult. I would like to see more detailed analysis as well as experimental comparison. (4) For the results in Table 2, I wonder how you compute the accuracy for the method 'Self-SL only'. I'm afraid that there is no classification module in the self-supervised learning framework.

Correctness: Yes

Clarity: Yes

Relation to Prior Work: Yes

Reproducibility: Yes

Additional Feedback: Please address my concerms in section of Weaknesses.

[Author Response · NeurIPS 2020]

Thank you for the insightful comments from all reviewers. Those are very helpful to improve our submission.

**Additional baselines [R1-A1]** We agree that [TaBERT & TabNet] and VIME have common concepts - the pretext task
for self-supervised learning (self-SL) is recovering masked data. However, there are two major differences as well: (1)
VIME utilizes another pretext task (binary mask vector estimation) for self-SL, (2) VIME utilizes the imputed data
as augmented samples for semi-SL. We include TaBERT and TabNet as additional baselines and the brief results (for
Table 2) are: [0.8653, 0.9463, 0.8127] and [0.8637, 0.9487, 0.8114] which are consistently worse than VIME.
**Extra ablation study [R1-A2]** These results are already presented in Section 8 in Appendix (see Table 4). The
combination of two pretext tasks is consistently better than only using one of those two pretext tasks.
**Correlation structure of tabular data [R1-A3]** The spatial correlations between pixels in images or the sequential
correlations between words in text data are well-known and consistent across different datasets. By contrast, the
correlation structure among features in tabular data is unknown and varies across different datasets. In other words,
there is no *common* correlation structure in tabular data.
**Novelties [R1-A4]** The design of VIME is dedicated to tabular data. The pretext tasks we use here mark a departure
from those used previously on image and text data. The main novelties of the VIME framework are (1) novel pretext
task(s) for tabular data (the combination of two pretext tasks) in Section 4.1 and (2) novel data augmentation for tabular
data in Section 4.2. We will clarify these novelties in the revised manuscript and tone down the semi-SL part.
**Alternative augmentation [R1-A5]** Thank you for the suggestion, though we would note that augmentation methods
for tabular data are not standardized. Note also that we included "MixUp" results as an additional augmentation model
in the manuscript (and it underperformed VIME in all experimental settings). We have since performed an additional
experiment in which we add Gaussian noise to the original data and treat this as the augmented sample. Briefly, as
compared with results in Table 2, performances with Gaussian noise augmentations are 0.8627, 0.9481, 0.8253 with
Income, MNIST, and Blog datasets, respectively. These results are consistently worse than the performances of VIME.
**Minor issues [R1-A6]** We will fix those typos and improve Figure 1 and 2 in the revised manuscript.

**Categorical variables [R2-A1]** We agree that categorical variable issues in tabular data is critical. In practice and
experiments, we change Eq (6) to cross-entropy loss for categorical features to properly handle them. We will clarify this.
Most of the tabular datasets used in our paper include categorical variables. For instance, all the variables in genomic
datasets are categorical. Also, two clinical datasets include various categorical features (see Table 2 in Appendix).
**PCA on Genomics [R2-A2]** We performed extra experiments using PCA + ElasticNet and PCA + Linear on genomics
data (as suggested). Unfortunately, the performance of PCA + ElasticNet and of PCA + Linear are consistently and
significantly worse than original ElasticNet and Linear models in terms of MSE. As explained in **R2-A1**, all the variables
in genomic data are categorical features; and usually, applying PCA on categorical variables is not recommended.
**Data normalization [R2-A3]** Yes, we did. We first use MinMaxScaler to normalize the data between 0 and 1 (it can
be also checked in the submitted codes (data-loader.py)); then, we train the self and semi-supervised models. We also
tried StandardScaler for data normalization (mean=0, std=1); and the performances were similar with MinMaxScaler.
**Clarification [R2-A4]** In the revised paper, we will clarify the meaning of "Variants of VIME" in the captions.

**Novelties [R3-A1]** We acknowledge that self/semi-supervised learning is well-studied in the image and language
domains. However, as shown in various results in the manuscript (e.g., Table 2) and Appendix (e.g., Table 5), the
state-of-the-art self/semi-supervised learning models for image and language domains (such as SimCLR) underperform
VIME in the tabular setting. This demonstrates that new self/semi-supervised learning models are necessary for tabular
domain (we also highlight our novelties in **R1-A4**). Note that VIME consistently outperforms Gaussian noise based
models in various settings not only shown in the manuscript (results with DAE baseline) but also proved in **R1-A5**.
**Masking the data [R3-A2]** The first term $(\mathbf{m} \cdot \bar{\mathbf{x}})$ consists precisely of the shuffled samples (with $\mathbf{m}$ determining which
features will be shuffled for the given data sample $\mathbf{x}$. Essentially the masked samples are partially masked (as is the
typical meaning of *masking*) - some of the sample consists of truly observed features, and the remainder is masked by
the *shuffling* you refer to. In this case, the marginal distributions of the corrupted samples are valid but conditional
distributions are invalid. Therefore, the encoder must consider the values of other components to estimate whether that
component is corrupted. In other words, the encoder should learn the *joint distributions* (which is the main objective of
self-supervised learning).
**Masking with Gaussian [R3-A3]** If we sample $\mathbf{m}$ with Gaussian distribution (instead of Bernoulli distribution), the
prediction performances for Income, MNIST, and Blog datasets are 0.8427, 0.9439, 0.8127 which are consistently
worse than VIME (See Table 2). Note that the performance degradation is significant with datasets including categorical
variables (Income and Blog). This is because, with Gaussian noise, the encoder can easily identify which feature is
corrupted if the corrupted features are categorical variables. Please see **R2-A1** and **R1-A5** for more details.
**Clarification [R3-A4]** To clarify the "Self-SL only" variant, it can be interpreted as $\beta = 0$. More specifically, we first
train the encoder via self-supervised learning. Then, we train the predictive model with loss function (in Eq 7) with
$\beta = 0$ (only utilizing the labeled data). We will clarify this in the revised manuscript.

[Meta-Review · NeurIPS 2020]

This paper proposes a new reconstruction loss for unsupervised training of representations. This loss extends auto-encoders via a pretext task that uses the marginal distribution of features. The reviewers were unanimous in their decision to accept this paper. (Note: the one remaining score of 5 was by a reviewer who wrote that after rebuttal they are raising their score to a 6).